# On the Vulnerability of Backdoor Defenses for Federated Learning

## Abstract

Federated learning (FL) is a distributed machine learning paradigm which enables jointly training a global model without sharing clients' data. However, its repetitive server-client communication gives room for possible backdoor attacks which misleads the global model into a targeted misprediction when a specific trigger pattern is presented. In response to such backdoor threats on federated learning, various defense measures have been proposed. In this paper, we study whether the current defense mechanisms truly neutralize the backdoor threats from federated learning in a practical setting by proposing a new federated backdoor attack framework for possible countermeasures. Different from traditional training (on triggered data) and rescaling (the malicious client model) based backdoor injection, the proposed backdoor attack framework (1) directly modifies (a small proportion of) local model weights to inject the backdoor trigger via sign flips; (2) jointly optimize the trigger pattern with the client model, thus is more persistent and stealthy for circumventing existing defenses. In a case study, we examine the strength and weaknesses of several recent federated backdoor defenses from three major categories and provide suggestions to the practitioners when training federated models in practice.

## 1 Introduction

In recent years, Federated Learning (FL) [22, 38] prevails as a new distributed machine learning paradigm, where many clients collaboratively train a global model without sharing clients' data. FL techniques have been widely applied to various real-world applications including keyword spotting [17], activity prediction on mobile devices [10, 36], smart sensing on edge devices [12], etc. Despite FL's collaborative training capability, it usually deals with heterogeneous (non-i.i.d.) data distribution among clients and its formulation naturally leads to repetitive synchronization between the server and the clients. This gives room for attacks from potential malicious clients. Particularly, backdoor attack [8], which aims to mislead the model into a targeted misprediction when a specific trigger pattern is presented by stealthy data poisoning, can be easily implemented and hard to detect from the server's perspective. The feasibility of backdoor attacks on plain federated learning has been studied in [3, 2, 35]. Such backdoor attacks can be effectively implemented by replacing the global FL model with the attackers' malicious model through carefully scaling model updates with well-designed triggers, and the attacks can successfully evade many different FL setups [22, 37].

The possible backdoor attacks in federated learning arouse a large number of interest on possible defenses that could mitigate the backdoor threats. Based on the different defense mechanisms they adopt, the federated backdoor defenses can be classified into three major categories: *model-refinement*, *robust-aggregation*, and *certified-robustness*. *Model-refinement* defenses attempt to refine the global model to erase the possible backdoor, through methods such as fine-tuning [33] or distillation [20, 29]. Intuitively, distillation or pruning-based FL can also be more robust to current federated backdoor

attacks as recent studies on backdoor defenses [19, 21] have shown that such methods are effective in removing backdoor from general (non-FL) backdoored models. On the other hand, different from FedAvg [22] and its variants [13, 18, 31], the *robust-aggregation* defenses exclude the malicious (ambiguous) weights and gradients from suspicious clients through anomaly detection, or dynamically re-weight clients' importance based on certain distance metrics (geometric median, etc.). Examples include Krum [4], Trimmed Mean [37], Bulyan [9] and Robust Learning Rate [24]. Note that some of the *robust-aggregation* defenses are originally proposed for defending model poisoning attack (Byzantine robustness) yet they may also be used for defending backdoors. The last kind, *certified robustness* aims at providing certified robustness guarantees that for each test example, i.e., the prediction would not change even some features in local training data of malicious clients have been modified within certain constraint. For example, CRFL [34] exploits clipping and smoothing on model parameters, which yields a sample-wise robustness certification with magnitude-limited backdoor trigger patterns. Provable FL [5] learns multiple global models, each with a random client subset and takes majority voting in prediction, which shows provably secure against malicious clients.

There exist many constraints and limitations for defenses from these three major categories. First, the effectiveness of the *model-refinement* defenses relies on whether the refinement procedure fully erase the backdoor. Adversaries may actually design robust backdoor patterns that are persistent, stealthy and thus hard to erase. Second, *robust aggregation* defenses are usually based on i.i.d. assumptions of each participant's training data, which does not hold for the general federated learning scenario where participant's data are usually non-i.i.d. Existing attacks [2] have shown that in certain cases, such defense techniques make the attack even more effective. Moreover, to effectively reduce the attack success rate of the possible backdoor attacks, one usually needs to enforce stronger robust aggregation rules, which can in turn largely hurt the normal federated training progress. Lastly, *certified robustness* approaches enjoy theoretical robust guarantees, yet also have strong requirements and limitations such as a large amount of model training or a strict limit on the magnitude of the trigger pattern. Also, certified defenses usually lead to relatively worse empirical model performances.

We propose a new federated backdoor attack that is more persistent and stealthy for circumventing most existing defenses. We summarize our main contributions and findings as follows:

- We propose a more persistent and stealthy federated backdoor attack. Instead of traditional training (on triggered data) and rescaling (the malicious client model) based backdoor injection, our attack selectively flips the signs of a small proportion of network weights and jointly optimizes the trigger pattern with the model.

- The proposed attack does not explicitly scale the updated weights (gradients) and can be universally applied to various architectures beyond convolutional neural networks, which is of independent interest to general backdoor attack and defense studies.

- We examine the effectiveness of recent federated backdoor defenses from three major categories and give practical guidelines for the choice of the backdoor defenses for different settings.

## 2 Proposed Approach

### 2.1 Preliminaries

**Federated Learning Setup** Suppose we have $K$ participating clients, each of which has its own dataset $\mathcal{D}_i$ with size $n_i$ and $N = \sum_i n_i$. At the $t$-th federated training round, the server send the current global model $\boldsymbol{\theta}_t$ to a randomly-selected subset of $m$ clients. The clients will then perform $K$ steps of local training to obtain $\boldsymbol{\theta}_t^{i,K}$ based on the global model $\boldsymbol{\theta}_t$, and send the updates $\boldsymbol{\theta}_t^{i,K} - \boldsymbol{\theta}_t$ back to the server. In the standard FedAvg [22] method, the server adopts a sample-weighted aggregation rule to average the $m$ received updates:

$$\boldsymbol{\theta}_{t+1} = \boldsymbol{\theta}_t + \frac{1}{N} \sum_{i=1}^{m} n_i(\boldsymbol{\theta}_t^{i,K} - \boldsymbol{\theta}_t) \tag{2.1}$$

**Backdoor Attacks in FL** Assume there exists one or several malicious clients with goal to manipulate local updates to inject a backdoor trigger into the global model such that when the trigger pattern appears in the inference stage, the global model would give preset target predictions $y_{\text{target}}$. In the meantime, the malicious clients do not want to tamper with the model's normal prediction accuracy on clean tasks (to keep stealthy). Therefore, the malicious client has the following objectives:

$$\min_{\boldsymbol{\theta}} \mathcal{L}_{\text{train}}(\mathbf{x}, \mathbf{x}', y_{\text{target}}, \boldsymbol{\theta}) := \frac{1}{n_i} \sum_{k=1}^{n_i} \ell(f_{\boldsymbol{\theta}}(\mathbf{x}_k), y_k) + \lambda \cdot \ell(f_{\boldsymbol{\theta}}(\mathbf{x}'_k), y_{\text{target}}) + \alpha \cdot ||\boldsymbol{\theta} - \boldsymbol{\theta}_{t-1}||_2^2, \quad (2.2)$$

where $\mathbf{x}'_k = (\mathbf{1} - \mathbf{m}) \odot \mathbf{x}_k + \mathbf{m} \odot \boldsymbol{\Delta}$ is the backdoored data and $\boldsymbol{\Delta}$ denotes the associated trigger pattern, $\mathbf{m}$ denotes the trigger location mask, and $\odot$ denotes the element-wise product. The first term in (2.2) is the common empirical risk minimization while the second term aims at injecting the backdoor trigger into the model. The third term is usually employed to enhance the attack stealthiness by minimizing the distance to the global model. $\lambda$ and $\alpha$ control the trade-off between the three tasks.

**Threat Model** We suppose that the malicious attackers have full control of their local training processes, such as backdoor data injection, trigger pattern, and local optimization. The scenario is practical since the server can only get the trained model from clients without the information on how the model is trained. Correspondingly, an malicious attacker is unable to influence the operations conducted on the central server such as changing the aggregation rules.

## 2.2 Focused Flip Federated Backdoor Attack

In this section, we propose **F**ocused-**F**lip **F**ederated **B**ackdoor **A**ttack (F3BA), in which the malicious clients only compromise a small fraction of the least important model parameters through *focused weight sign manipulation*. The goal of such weight sign manipulation is to cause a strong activation difference in each layer led by the presence of the trigger pattern while keeping the modification footprint and influence on model accuracy minimal. A sketch of our proposed attack is illustrated in Figure 1. Let's denote the current global model as $\boldsymbol{\theta}_t^{i,0} := \{\mathbf{w}_t^{[1]}, \mathbf{w}_t^{[2]}, .., \mathbf{w}_t^{[L]}\}$ and each layer's output as $\mathbf{z}^{[1]}(\cdot), \mathbf{z}^{[2]}(\cdot), .., \mathbf{z}^{[L]}(\cdot)$. Generally, our attack can be divided into three steps:

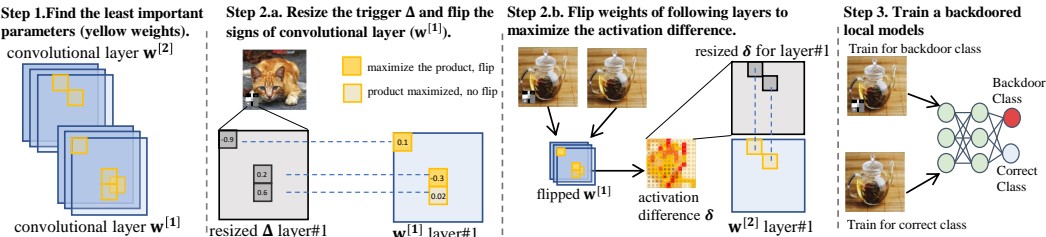

Figure 1: A sketch of our proposed Focused Flip Federated Backdoor Attack.

**Step 1: Search candidate parameters for manipulation.** We only manipulate a small fraction of candidate parameters in the model that are of the least importance to the normal task to make it have a slight impact on the natural accuracy. Specifically, we introduce *movement-based* importance score to identify candidate parameters for manipulation, which is inspired by the movement pruning [27]. Specifically, the importance of each parameter $\boldsymbol{S}_t^{[j]}$ is related to both its weight and gradient: $\boldsymbol{S}_t^{[j]} = -\frac{\partial \mathcal{L}_g}{\partial \mathbf{w}_t^{[j]}} \odot \mathbf{w}_t^{[j]}$, where $\mathcal{L}_g$ is the global training loss and $\odot$ denotes the elementwise product[1]. We make two major changes on $\boldsymbol{S}_t^{[j]}$ for our federated backdoor attack:

- In our federated setting, it is hard to obtain the global loss $\mathcal{L}_g$ since the attack is carried only on the malicious workers. We simply approximate the partial derivative with the model difference $-\frac{\partial \mathcal{L}_g}{\partial \mathbf{w}_t^{[j]}} \approx \mathbf{w}_t^{[j]} - \mathbf{w}_{t-1}^{[j]}$. When $t = 0$ we simply randomly generate the importance score[2] $\boldsymbol{S}_0^{[j]}$;

- To handle defense mechanisms with different emphasizes, we extend it into two importance metrics and choose[3] the one that best exploits the weakness of the defense:

$$\text{Directional Criteria: } \boldsymbol{S}_t^{[j]} = (\mathbf{w}_t^{[j]} - \mathbf{w}_{t-1}^{[j]}) \odot \mathbf{w}_t^{[j]}, \quad (2.3)$$

$$\text{Directionless Criteria: } \boldsymbol{S}_t^{[j]} = \left|(\mathbf{w}_t^{[j]} - \mathbf{w}_{t-1}^{[j]}) \odot \mathbf{w}_t^{[j]}\right|. \quad (2.4)$$

Given the importance score $\boldsymbol{S}_t^{[j]}$, we choose the least important parameters in each layer as candidate parameters. We define $\mathbf{m}_s^{[j]}$ as a mask that selects the $s\%$ lowest scores in $\boldsymbol{S}_t^{[j]}$ and ignore the others. In practice, setting $s = 1\%$ for the total model parameters is usually sufficient for our attack.

---

[1] More explanations regarding this movement-based importance score can be found in the Appendix.

[2] In practice, since only a subset of clients participate in the training procedure, the malicious client keeps its last received global model until next time it is chosen for training and compute the model difference term.

[3] A detailed discussion on how to choose the appropriate criteria can be found in the Appendix.

**Step 2: Flip the sign of candidate parameters**: We manipulate the parameters to enhance their sensitivity to the trigger by flipping their signs. Take the simple CNN model as an example[4]. We start flipping from the first convolutional layer. For a given trigger pattern[5] $\mathbf{\Delta}$, to maximize the activation in the next layer, we flip $\mathbf{w}^{[1]}$'s signs if they are different from the trigger's signs in the same position:

$$\mathbf{w}^{[1]} = \mathbf{m}_s^{[1]} \odot \text{sign}(\mathbf{\Delta}) \odot |\mathbf{w}^{[1]}| + (\mathbf{1} - \mathbf{m}_s^{[1]}) \odot \mathbf{w}^{[1]}, \tag{2.5}$$

where $\mathbf{m}_s^{[1]}$ is the candidate parameter mask generated in Step 1. Through (2.5), the activation in the next layer is indeed enlarged when the trigger pattern is present. For the subsequent layers, we flip the signs of the candidate parameters similarly. The only difference is that after the sign-flip in the previous layer $j - 1$, we feed a small set of validation samples $\mathbf{x}_v$ and compute the activation difference of layer $j - 1$ caused by adding the trigger pattern $\mathbf{\Delta}$ on $\mathbf{x}_v$:

$$\boldsymbol{\delta} = \sigma(\mathbf{z}^{[j-1]}(\mathbf{x}_v')) - \sigma(\mathbf{z}^{[j-1]}(\mathbf{x}_v)), \text{ where } \mathbf{x}_v' = (\mathbf{1} - \mathbf{m}) \odot \mathbf{x}_v + \mathbf{m} \odot \mathbf{\Delta}$$

$\sigma(\cdot)$ is the activation function for the network (e.g. ReLU function) and $\mathbf{x}_v'$ is the backdoor triggered validation samples. Similarly, we can flip the signs of the candidate parameters to maximize $\boldsymbol{\delta}$. This ensures that the last layer's activation is also maximized when the trigger pattern is presented.

**Step 3: Model training**: Although we have maximized the network's activation for the backdoor trigger in Step 2, the local model training step is still necessary due to: 1) the flipped parameters only maximize the activation but have not associated with the target label $y_{\text{target}}$, and the training step using (2.2) would bind the trigger to the target label; 2) flipping the signs of the parameter will lead to a quite different model update compared with other benign clients and a further training step largely mitigates this issue. Broadly speaking, Focused Flip greatly boosts training-based backdoor attacks, whereas its time overhead is negligible as the flipping operation does not depend on backpropagation.

## 2.3 Optimize the trigger pattern

To further improve the effectiveness of our proposed F3BA, we equip the attack with trigger pattern optimization[6], i.e., instead of using a fixed trigger, we optimize the trigger to fit our attack. Specifically, trigger optimization happens in the middle of Step 2 and repeats for $P$ iterations: in each iteration, we first conduct the same focused-flip procedure for $\mathbf{w}^{[1]}$. Then we draw batches of training data $\mathbf{x}_p$ and generate the corresponding triggered data $\mathbf{x}_p'$ using the current trigger $\mathbf{\Delta}$. We feed both the clean samples $\mathbf{x}_p$ and the triggered samples $\mathbf{x}_p'$ to the first layer and design the trigger optimization loss to maximize the activation difference:

$$\max_{\mathbf{\Delta}} \mathcal{L}_{\text{trig}}(\mathbf{x}_p, \mathbf{\Delta}) := \|\sigma(\mathbf{z}^{[1]}(\mathbf{x}_p) - \sigma(\mathbf{z}^{[1]}(\mathbf{x}_p'))\|_2^2, \text{ where } \mathbf{x}_p' = (\mathbf{1} - \mathbf{m}) \odot \mathbf{x}_p + \mathbf{m} \odot \mathbf{\Delta}.$$

In practice, we optimize $\mathcal{L}_{\text{trig}}$ via simple gradient ascent. It is noteworthy that since the pattern $\mathbf{\Delta}$ is being optimized in each iteration, we need to re-flip the candidate parameters in $\mathbf{w}^{[1]}$ to follow such changes. The remaining steps for flipping the following layers are the same as before.

# 3 Evaluating the State-of-the-Art Federated Backdoor Defenses

We evaluate F3BA with trigger optimization on several state-of-the-art federated backdoor defenses (3 model-refinement defenses, 3 robust-aggregation defenses, and 1 certified defense) and compare with the distributed backdoor attack (DBA) [35]. We test on CIFAR-10 [15] and Tiny-ImageNet [16] with a plain CNN and Resnet-18 model respectively under the non-i.i.d. data distributions. The performances of the federated backdoor attacks is measured by two metrics: Attack Success Rate (ASR), i.e., the proportion of the triggered samples classified as target labels and Natural Accuracy (ACC), i.e., prediction accuracy on the natural clean examples. We test the global model after each round of aggregation: we use the clean test dataset to evaluate ACC, average all the optimized triggers as a global trigger and attached it to the test dataset for ASR evaluation..

## 3.1 Attacking Model-Refinement Defenses

**FedDF** [20] performs server-side model fusion, i.e. distill the next round global model using the outputs of all the clients' models on the unlabeled data. Specifically, FedDF ensembles all the client models $\boldsymbol{\theta}_t^{i,K}$ together as the teacher model, and use it to distill the next round global model.

---

[4]Note that it also applies to fully connected layers with simple modifications.

[5]If the size of the trigger is not aligned with $\mathbf{w}^{[1]}$, we simply resize it into the same size as $\mathbf{w}^{[1]}$

[6]The detailed algorithm for trigger optimization can be found in the Appendix.

167 **FedRAD** [29] defends by giving each client a median-
168 based score, which measures the frequency that the
169 client output logits become the median for class pre-
170 dictions. The distillation part is similar to FedDF.

171 **FedMV Pruning** [33] is a distributed pruning scheme
172 to mitigate backdoor attacks in FL, where each client
173 provides ranks for all filters in the last convolutional
174 layer based on their activation values on local test sam-
175 ples. The server averages the received rankings, and
176 prunes the filters of the global model's last convolu-
177 tional layer with large averaged rankings. Besides,
178 FedMV Pruning erases the outlier weights (far from
179 the average parameter weight) every few rounds.

180 **Results**: From Figure 2, both DBA and F3BA pene-
181 trate the three defenses on the CIFAR-10 with closed
182 ACC. On the Tiny-ImageNet dataset, the DBA's ASR
183 soon decreases as the training proceeds, suggesting
184 the benign updates overpower the malicious ones and
185 dominate in global updates. F3BA still evades all
186 three defenses with higher accuracy. Standalone from
187 ensemble distillation, FedMV pruning causes sudden
188 ACC loss in some rounds due to setting some weights
189 with large magnitudes to zero, and these weights can
190 be important to the main task.

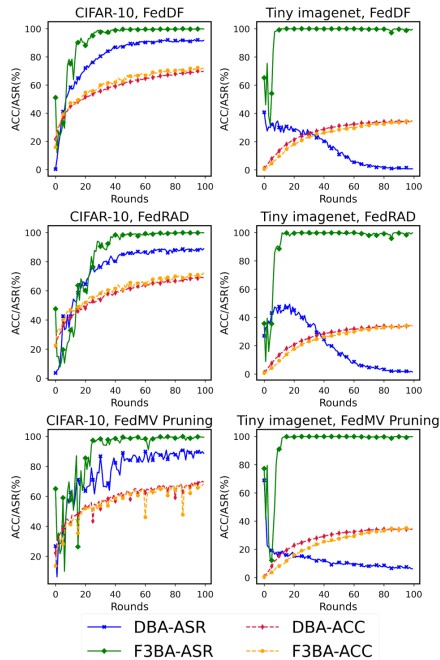

Figure 2: ASR and ACC of F3BA and DBA against Model-Refinement Defenses.

### 3.2 Attacking Robust-Aggregation Defenses

192 **Bulyan** [9] is one of the strongest Byzantine-resilient robust aggregation algorithms originally
193 designed for model poisoning attacks.

194 It works by ensuring that each coordinate is agreed on
195 by a majority of vectors selected by a Byzantine resilient
196 aggregation rule. It requires that for each aggregation,
197 the total number of clients $n$ satisfy $n \geq 4f + 3$, $f$ is
198 the number of malicious clients. To efficiently evade
199 Bulyan, we replace directional criteria (eq. (2.3)) with
200 directionless criteria (eq. (2.4)) to find candidate pa-
201 rameters with small magnitudes and updates[7] **Robust**
202 **LR** [24] adjusts the servers' learning rate based on the
203 sign of clients' updates: it requires a sufficient number
204 of votes on the signs of the update for each dimension
205 to move towards a certain direction. For dimensions
206 where the sum of signs is below the threshold, Robust
207 LR maximizes the loss. For other dimensions, it tries to
208 minimize the loss as usual.

209 **DeepSight** [26] aims to filter malicious clients to miti-
210 gate the backdoor: it clusters clients with different met-
211 rics and removes the cluster whose identified malicious
212 clients exceeds a threshold. Specifically, 1) it inspects
213 the output probabilities of local models on random inputs
214 $\mathbf{x}_{\text{rand}}$ to decide whether its training samples concentrate
215 on a particular class (likely backdoors); 2) it applies
216 DBSCAN [6] to cluster clients and excludes the entire
217 cluster if the number of potentially malicious clients
218 exceeds a threshold.

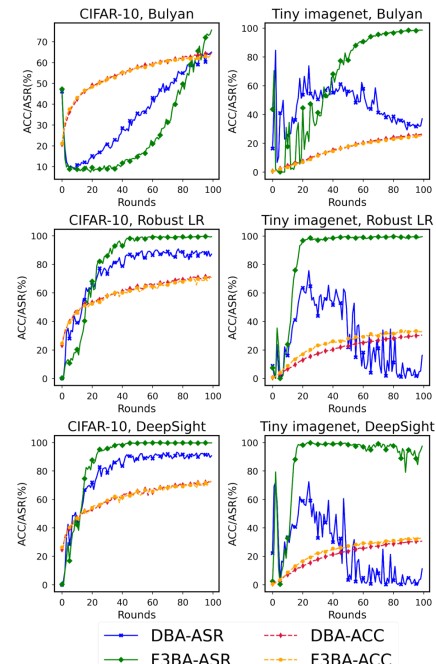

Figure 3: ASR and ACC of F3BA and DBA against Robust Aggregation Defenses.

---

[7]The reason of this choice are discussed in the Appendix.

219 **Results**: Bulyan's iterative exclusion of anomaly updates undermines the evasion of F3BA. By
220 applying directionless criteria when flipping parameters, F3BA boosts its stealthiness and achieves
221 high ASR. For Robust LR, Bulyan exploits the restriction of its voting mechanism and hacks into it.
222 DeepSight can not defend F3BA under the extremely non-i.i.d data distribution either. In comparison,
223 DBA fails on Tiny-Imagenet Dataset under all three defenses under the same client numbers and data
224 heterogeneity.

### 3.3  Attacking Certified-robustness

226 **CRFL** [34] gives each sample a certificated radius $RAD$ that the prediction would not change if (part
227 of) local training data is modified with backdoor magnitude $||\mathbf{\Delta}|| < RAD$. It provides robustness
228 guarantee through clipping and perturbing in training and parameter-smoothed testing.

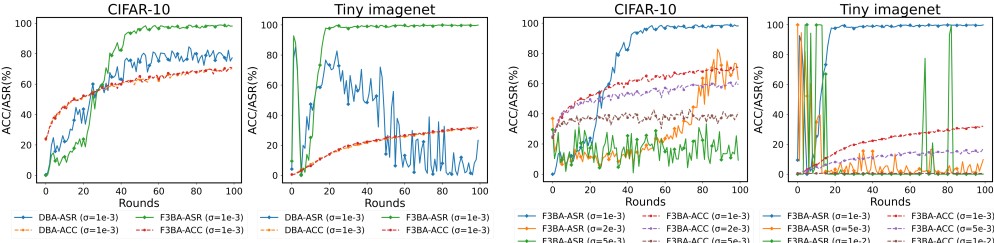

Figure 4: ACC and ASR of F3BA and DBA against CRFL with $\sigma = 0.001$.

Figure 5: ACC and ASR of F3BA against CRFL with different $\sigma$.

229 **Results**: To test the defense performance of CRFL, we adjust the variance $\sigma$ for CRFL and test with
230 backdoor attacks. Figure 4 shows that using the same level of noise $\sigma = 0.001$, F3BA reaches the
231 ASR of nearly 100% and DBA fails on Tiny-Imagenet. If we further increase variance to provide
232 larger RAD for F3BA that completely covers the norm of the trigger in each round as in Figure 5,
233 CRFL can defend the F3BA yet with a huge sacrifice on accuracy.

## 4  Takeaway for Practitioners

235 From the results in Section 3, current federated backdoor defenses, represented by the three categories,
236 all have their own Achilles' heel facing stealthier and more adaptive attacks such as F3BA: *model-*
237 *refinement* defenses enhance the global model's robustness towards data drift while fail to erase the
238 backdoor in malicious updates; certain *robust-aggregation* (e.g., Bulyan, Robust LR) and *certified-*
239 *robustness* (e.g., CRFL) defenses achieve acceptable backdoor defense capabilities when imposing
240 strong intervention mechanisms such as introducing large random noise or reversing global updates.
241 However, such strong interventions also inevitably hurt the model's natural accuracy. Overall, we
242 recommend the practitioners to adopt Bulyan or CRFL in the cases where the natural accuracy is
243 already satisfiable or is less important, as they are the most helpful in defending against backdoors.

## 5  Conclusions

245 In this paper, we propose F3BA to backdoor federated learning. Our attack does not require explicitly
246 scaling malicious uploaded clients' local updates but instead flips the weights of some unimportant
247 model parameters for the main task. With F3BA, we evaluate the current state-of-the-art backdoor
248 defenses in federated learning. In most of the tests, F3BA is able to evade and reach a high attack
249 success rate. From this we argue that despite providing some robustness, the current stage of backdoor
250 defenses still expose the vulnerability to the advanced backdoor attacks.

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

## A    Additional Related Work

There are a large body of works on federated learning. In this section, we only review the most relevant works in general FL as well as the backdoor attack and defenses of federated learning.

**Federated Learning**: Federated Learning [14] was proposed for improving the communication efficiency in distributed settings. FedAvg [22] works by averaging local SGD updates, of which the variants have also been proposed such as SCAFFOLD [13], FedProx [18], FedNova [31]. [25, 32] proposed adaptive federated optimization methods for better adaptivity. Recently, new aggregation strategies such as neuron alignment [28] or ensemble distillation [20] has also been proposed.

**Backdoor Attacks on Federated Learning**: [2] injects backdoor by predicting the global model updates and replacing them with the one that was embedded with backdoors. [3] aims to achieve both global model convergence and targeted poisoning attack by explicitly boosting the malicious updates and alternatively minimizing backdoor objectives and the stealth metric. [30] shows that robustness to backdoors implies model robustness to adversarial examples and proposed edge-case backdoors. DBA [35] decomposes the trigger pattern into sub-patterns and distributing them for several malicious clients to implant. With known aggregation rules, malicious clients also speculate the global update direction and modify it for the specified direction for backdoor task [7].

**Backdoor Defenses on Federated Learning**: In federated learning, the server does not have access to the clients' local data, so some defenses based on data inspection (removing training data with backdoors) [11] are not applicable. Robust Learning Rate [24] flips the signs of some dimensions of global updates. [33] designs a federated pruning method to remove redundant neurons for backdoor defense. [34] proposed a certified defense that exploits clipping and smoothing for better model smoothness. BAFFLE [1] uses a set of validating clients, refreshed in each training round, to determine whether the global updates have been subject to a backdoor injection. Recent work [26] identifies suspicious model updates via clustering-based similarity estimations.

## B    Extensions to Other Network Architectures

The flip operation can be similarly extended to other network architectures as the candidate weights selection (Step 1) and the model training (Step 3) are not relevant to the model architecture at all. Therefore, we only need to adapt the sign flipping part (Step 2). For CNN, we resize the trigger and flip the sign of the candidate parameters to maximize the convolution layer's activation. The same strategy applies for any dot product based operation (convolution can be seen as a special dot product). Take MLP as an example, assume the first layer's weight is $\mathbf{w}^{[1]}$. We flip these weights' signs by $\mathbf{w}^{[1]} = \mathbf{m}_s^{[1]} \odot \text{sign}(\mathbf{x}'_{\text{in}} - \mathbf{x}_{\text{in}}) \odot |\mathbf{w}^{[1]}| + (\mathbf{1} - \mathbf{m}_s^{[1]}) \odot \mathbf{w}^{[1]}$, ($\mathbf{x}'_{\text{in}}$ and $\mathbf{x}_{\text{in}}$ is the flatten input sample with and without trigger, and the non-zero elements of $\mathbf{x}'_{\text{in}} - \mathbf{x}_{\text{in}}$ only take place on pixels with the trigger.) The Equation is similar to Equation 2.5 except that we do not need to resize the trigger as in CNN. The sign flipping of the rest layers follows the same.

## C    Algorithms

### C.1    Focused-Flip Federated Backdoor Attack

We summarize our F3BA algorithm as pseudo-code in Algorithm 1. Specifically, our F3BA method searches for candidate parameters (Line 6 - Line 9) based on directional or directionless criteria, such that it is difficult to remove these backdoor-related parameters by coarse-scale model-refinement or

aggregation rules on the server. After each time flipping candidate parameters in a certain layer, the same validation samples $\mathbf{x}_v$ are required (Line 12) to calculate the activation difference (Line 15) for flipping candidate parameters in the following layers.

---

**Algorithm 1** Focused-Flip Federated Backdoor Attack (F3BA)

---

1: **Input:** $\sigma(\cdot)$: activation function
2: $\quad\quad\quad$ $\alpha$: learning rate for backdoor training
3: $\quad\quad\quad$ $K$: local training iterations
4: $\quad\quad\quad$ $\mathcal{L}_{\text{train}}$: loss function for backdoor training
5: Malicious client $i$ receives $\boldsymbol{\theta}_t^{i,0} := \{\mathbf{w}^{[1]}, \mathbf{w}^{[2]}, .., \mathbf{w}^{[L]}\}$ from the server. Let us denote each layer's output as $\mathbf{z}^{[1]}(\cdot), \mathbf{z}^{[2]}(\cdot), .., \mathbf{z}^{[L]}(\cdot)$.
6: **for** $j = 1$ **to** $L$ **do**
7: $\quad$ Compute $\boldsymbol{S}_t^{[j]}$ based on Equation (2.3) or Equation (2.4)
8: $\quad$ Compute $\mathbf{m}_s^{[j]}$ as a mask that selects $s\%$ lowest scores in $\boldsymbol{S}_t^{[j]}$
9: **end for**
10: Resize $\boldsymbol{\Delta}$ to the size of $\mathbf{w}^{[1]}$ and get $\boldsymbol{\Delta}^*$
11: $\mathbf{w}^{[1]} = \mathbf{m}_s^{[1]} \odot \text{sign}(\boldsymbol{\Delta}^*) \odot |\mathbf{w}^{[1]}| + (\mathbf{1} - \mathbf{m}_s^{[1]}) \odot \mathbf{w}^{[1]}$
12: Sample a batch of validation data $\mathbf{x}_v$ from $\mathcal{D}_i$
13: **for** $j = 2$ **to** $L$ **do**
14: $\quad$ $\mathbf{x}_v' = (\mathbf{1} - \mathbf{m}) \odot \mathbf{x}_v + \mathbf{m} \odot \boldsymbol{\Delta}$ //backdoor generation
15: $\quad$ $\boldsymbol{\delta} = \sigma(\mathbf{z}^{[j-1]}(\mathbf{x}_v')) - \sigma(\mathbf{z}^{[j-1]}(\mathbf{x}_v))$
16: $\quad$ Resize $\boldsymbol{\delta}$ to the size of $\mathbf{w}^{[j]}$ and get $\boldsymbol{\delta}^*$
17: $\quad$ $\mathbf{w}^{[j]} = \mathbf{m}_s^{[i]} \odot \text{sign}(\boldsymbol{\delta}^*) \odot |\mathbf{w}^{[j]}| + (\mathbf{1} - \mathbf{m}_s^{[j]}) \odot \mathbf{w}^{[j]}$
18: **end for**
19: **for** $k = 1$ **to** $K$ **do**
20: $\quad$ Sample a batch of training data $\mathbf{x}_k$ from $\mathcal{D}_i$
21: $\quad$ $\mathbf{x}_k' = (\mathbf{1} - \mathbf{m}) \odot \mathbf{x}_k + \mathbf{m} \odot \boldsymbol{\Delta}$ //backdoor generation
22: $\quad$ $\boldsymbol{\theta}_t^{i,k} = \boldsymbol{\theta}_t^{i,k-1} - \alpha \nabla_{\boldsymbol{\theta}} \mathcal{L}_{\text{train}}(\mathbf{x}_k, \mathbf{x}_k', y_{\text{target}}, \boldsymbol{\theta}_t^{i,k-1})$
23: **end for**

---

## C.2 Focused-Flip Federated Backdoor Attack With Trigger Optimization

We also summarize the more advanced F3BA with trigger pattern optimization in Algorithm 2. The major difference compared with Algorithm 1 lies in the trigger optimization part (Line 14 - Line 19), where the first layer is repetitively flipped (Line 17) based on the signs of trigger pattern in the same position after each optimization step. After the trigger is optimized, the focused flip of the first layer is also finished, and the rest part (flipping the following layers) is the same as Algorithm 2.

## D Discussion on Selection Criterion of the Candidate Parameters

As shown in Eq. (2.3) and Eq. (2.4), we have two possible criteria (Directional or Directionless) for the selection of candidate parameters. In our paper, we set the Directional Criteria as the default setting for F3BA. Though it works well in most cases such as when attacking model-refinement defenses, we find that for some robust aggregation defenses (e.g., Bulyan), we need to adjust it to fit better. In this section, we discuss these two criteria: what are the meanings of the two criteria and how to pick the right one to use in different situations.

We first talk about the *Directional Criteria*, which is our default setting.

**Directional Criteria** target parameters that are moving significantly far away from 0 (and consider that as important [8] weight). To see this, note that we will obtain a large importance score under two scenarios: (1) when the $r$-th element in $\mathbf{w}_t^{[j]}$ is increasing, i.e., $\left[\frac{\partial \mathcal{L}_{\text{g}}}{\partial \mathbf{w}_t^{[j]}}\right]_r < 0$, and $\left[\mathbf{w}_t^{[j]}\right]_r > 0$; (2) when the $r$-th element in $\mathbf{w}_t^{[j]}$ is decreasing, i.e., $\left[\frac{\partial \mathcal{L}_{\text{g}}}{\partial \mathbf{w}_t^{[j]}}\right]_r > 0$, and $\left[\mathbf{w}_t^{[j]}\right]_r < 0$. Both scenarios

---

[8]Assume that we train the model from scratch, i.e., a model with all zero parameters. All important weights that significantly affect model accuracy will eventually move to either positive values or negative values.

**Algorithm 2** Focused-Flip Federated Backdoor Attack with Trigger Optimization (F3BA-TrigOpt)

1: **Input:** $\sigma(\cdot)$: activation function
2:         $\alpha$: learning rate for backdoor training
3:         $\eta$: learning rate for trigger optimization
4:         $K$: local training iterations
5:         $P$: trigger optimization iterations
6:         $\mathcal{L}_{\text{train}}$: loss function for backdoor training
7:         $\mathcal{L}_{\text{trig}}$: loss function for trigger optimization
8: Malicious client $i$ receives $\boldsymbol{\theta}_t^{i,0} := \{\mathbf{w}^{[1]}, \mathbf{w}^{[2]}, .., \mathbf{w}^{[L]}\}$ from the server. Let us denote each layer's output as $\mathbf{z}^{[1]}(\cdot), \mathbf{z}^{[2]}(\cdot), .., \mathbf{z}^{[L]}(\cdot)$.
9: **for** $j = 1$ **to** $L$ **do**
10:      Compute $\boldsymbol{S}_t^{[j]}$ based on Equation (2.3) or Equation (2.4)
11:      Compute $\mathbf{m}_s^{[j]}$ as a mask that selects $s\%$ lowest scores in $\boldsymbol{S}_t^{[j]}$
12: **end for**
13: **for** $p = 1$ **to** $P$ **do**
14:      Resize $\boldsymbol{\Delta}$ to the size of $\mathbf{w}^{[1]}$ and get $\boldsymbol{\Delta}^*$
15:      $\mathbf{w}^{[1]} = \mathbf{m}_s^{[1]} \odot \text{sign}(\boldsymbol{\Delta}^*) \odot |\mathbf{w}^{[1]}| + (\mathbf{1} - \mathbf{m}_s^{[1]}) \odot \mathbf{w}^{[1]}$
16:      Sample a batch of training data $\mathbf{x}_p$ from $\mathcal{D}_i$
17:      $\mathbf{x}_p' = (\mathbf{1} - \mathbf{m}) \odot \mathbf{x}_p + \mathbf{m} \odot \boldsymbol{\Delta}$ //backdoor generation
18:      $\mathcal{L}_{\text{trig}}(\mathbf{x}_p, \mathbf{x}_p') = \|\sigma(\mathbf{z}^{[1]}(\mathbf{x}_p)) - \sigma(\mathbf{z}^{[1]}(\mathbf{x}_p'))\|_2^2$
19:      $\boldsymbol{\Delta} = \boldsymbol{\Delta} + \eta \cdot \nabla_{\boldsymbol{\Delta}} \mathcal{L}_{\text{trig}}(\mathbf{x}_p, \mathbf{x}_p')$
20: **end for**
21: **for** $j = 2$ **to** $L$ **do**
22:      Sample a batch of validation data $\mathbf{x}_v$ from $\mathcal{D}_i$
23:      $\mathbf{x}_v' = (\mathbf{1} - \mathbf{m}) \odot \mathbf{x}_v + \mathbf{m} \odot \boldsymbol{\Delta}$ //backdoor generation
24:      $\boldsymbol{\delta} = \sigma(\mathbf{z}^{[j-1]}(\mathbf{x}_v)) - \sigma(\mathbf{z}^{[j-1]}(\mathbf{x}_v'))$
25:      Resize $\boldsymbol{\delta}$ to the size of $\mathbf{w}^{[j]}$ and get $\boldsymbol{\delta}^*$
26:      $\mathbf{w}^{[j]} = \mathbf{m}_s^{[j]} \odot \text{sign}(\boldsymbol{\delta}^*) \odot |\mathbf{w}^{[j]}| + (\mathbf{1} - \mathbf{m}_s^{[j]}) \odot \mathbf{w}^{[j]}$
27: **end for**
28: **for** $k = 1$ **to** $K$ **do**
29:      Sample a batch of training data $\mathbf{x}_k$ from $\mathcal{D}_i$
30:      $\mathbf{x}_k' = (\mathbf{1} - \mathbf{m}) \odot \mathbf{x}_k + \mathbf{m} \odot \boldsymbol{\Delta}$ //backdoor generation
31:      $\boldsymbol{\theta}_t^{i,k} = \boldsymbol{\theta}_t^{i,k-1} - \alpha \nabla_{\boldsymbol{\theta}} \mathcal{L}_{\text{train}}(\mathbf{x}_k, \mathbf{x}_k', y_{\text{target}}, \boldsymbol{\theta}_t^{i,k-1})$
32: **end for**

---

suggest that $\left[\mathbf{w}_t^{[j]}\right]_r$ is moving away from 0. On the contrary, when the parameter weight and its derivative is the same sign, the criterion regards this parameter as not important for its main task. F3BA exploits these unimportant parameters (moving towards 0) and flips their signs to mount a strong and persistent attack without damaging performance on the main task.

Despite having little influence on the main task, this criterion tends to select parameters with the largest absolute values on weights or updates (approximations to the derivatives) as candidates. Generally, it helps backdoor FL systems where a low proportion of malicious clients need to compete with a large number of benign ones, but also be defended by robust aggregation methods that filter extreme weights (updates). Therefore, we also propose the *Directionless Criteria* for such situations.

**Directionless Criteria** target parameters with the smallest magnitudes on both their parameter weights and updates. When a parameter's weight (update) is closer to 0 compared with other parameters, it would less likely to be regarded as an outlier or potentially malicious update. When the data are non-i.i.d distributed, the proposed local updates can not reach an agreement on the signs of some coordinates. In this circumstance, smaller $|\mathbf{w}_t^{[j]}|$ and smaller $|\frac{\partial \mathcal{L}_g}{\partial \mathbf{w}_t^{[j]}}|$ separately ensure that the candidate parameters would not be too large or small among all proposed ones before and after being flipped, ensuring stealth of the attack.

In summary, as a complement to directional criteria, directionless criteria bypass the robust aggregation defenses based on filtering or changing the extreme values of model parameters.

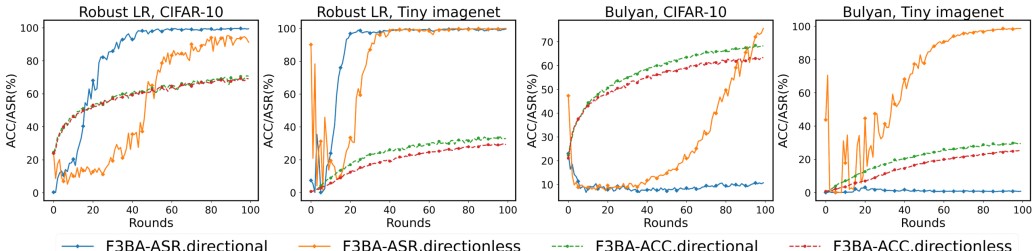

Figure 6: Attack success rate and accuracy against Robust LR and Bulyan on CIFAR-10 and Tiny imagenet datasets with directional and directionless criteria. 20 clients, 4 malicious.

In Figure 6, we show two examples: Robust LR and Bulyan, under F3BA attack using directional and directionless criteria respectively. From Figure 6, we can observe that using directional criteria in Robust LR helps reach a higher ASR in fewer rounds compared with directionless criteria while using the same directional criteria in Bulyan directly leads to failure of attack (Directionless Criteria is needed here). We argue that the gap in attack effectiveness with two criteria may be attributed to the defense mechanism. For Robust LR, since it does not put constraints on a single proposed update but the signs of all proposed ones, we can use directional criteria to reach a stronger attack. For Bulyan, however, the directional criterion makes the updates of flipped parameters become larger, with a higher probability to be excluded from the aggregation. To keep the malicious updates stealthy after being flipped, we turn to apply the directionless criteria.

## E    Additional Experimental Details

### E.1    General Experimental Settings

We evaluate the attacks on two classification datasets with non-i.i.d. data distributions: CIFAR-10 [15] and Tiny-ImageNet [16]. To simulate non-i.i.d. training data and supply the server with unbalanced samples from each class for model refinement, we divide the training images (in both datasets) using a Dirichlet distribution [23] with a concentration hyperparameter $h$. A shared global model is trained by all participants each round for aggregation. We evaluate CIFAR-10 and Tiny Imagenet dataset separately with a simple CNN (2 convolutional layers and 2 fully connected layers) and Resnet-18. Each participating client selected in one round will train for local epochs using SGD with the learning rate of $\eta = 0.001$ for both CIFAR and Tiny ImageNet. To ensure that the backdoor trigger is practical and hard to notice by human eyes, we limit the size of the trigger to a small $3 \times 3$ square on the CIFAR-10 dataset, and $4 \times 4$ for the Tiny ImageNet dataset. For F3BA, we set the trigger optimization iteration $P = 10$ and $\eta = 0.1$. We apply different candidate parameter selection porportions $1\%$ and $0.1\%$ respectively for convolutional layers and fully-connected layers.

### E.2    Discussion On Experimental Settings

We first test the effect of our proposed attack on plain FedAvg [22] without any defensive measures. Specifically, we compare our complete F3BA attack with trigger optimization in Algorithm 2 with a recent proposed, distributed backdoor attack (DBA) [35]. The main idea of DBA is to partition the trigger into multiple parts and distribute the backdoor injection task to several malicious clients by these sub-triggers. This ensures that the resulting malicious updates are less noticeable and more stealthy compared to standard federated backdoor attacks. We follow the same setting (e.g. number of total and malicious clients, local training epochs, learning rate $\eta$, concentration hyperparameter $h$) as the experiment of DBA for the attack of plain FedAvg.

As Figure 7, both F3BA and DBA can easily break FedAvg, and F3BA reaches a higher ASR without noticeable ACC loss. Though quickly observe the difference between centralized and distributed attacks, we argue that the DBA's experimental setting strongly facilitate the attack of DBA: all malicious clients are consistently selected in the total 100 clients and clients' data is very unevenly distributed (the concentration hyper-parameter $h$ is 0.1). Benign participants are randomly selected to form a total of selected 10 participants in each round. The malicious clients train 5 local epochs

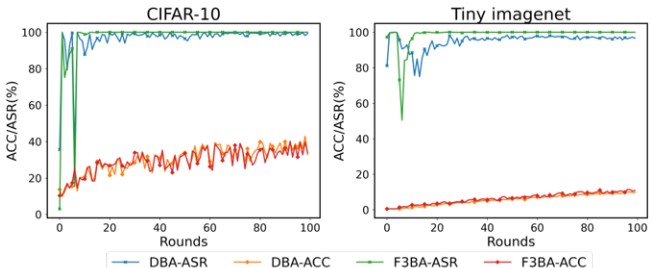

Figure 7: Attack success rate and accuracy of F3BA and DBA against plain FedAvg on CIFAR-10 and Tiny-ImageNet datasets under the setting of DBA experiment.

468 (including both clean and backdoored samples), more than 2 in benign clients. These settings hinder
469 the accumulation of benign updates thus making them unable to compete with the malicious updates.
470 We put our control experiments on these elements and the corresponding adjustments in Section **??**
471 and the attack settings of Section 3 to give a preciser and more fair evaluation on the effectiveness of
472 backdoor defences.

### E.3    Additional Details on the Defense Baselines

474 **FedDF** [20] leverages unlabeled data or artificially generated samples from a GAN's generator to
475 achieve robust server-side model fusion, aggregate knowledge from all received (heterogeneous)
476 client models. Formally, in the $t$-th round, the server first obtains a global model $\boldsymbol{\theta}_{t+1}$ with standard
477 FedAvg as equation 2.1.

478 FedDF distills the global model with the ensembled logit predictions on clean unlabeled samples from
479 each client's model, hence overcoming the limitation of the number of data samples and eliminating
480 the effect of data drift.

481 **FedRAD** [29] utilizes median-based scoring along with knowledge distillation for ensemble dis-
482 tillation. In the $t$-th round with $K$ participating clients, the median-based scoring assigns the $i$-th
483 client a score $s_i$ by counting how many times it gave the prediction for a server-side dataset $\mathbf{x}_s$. For
484 a classification task with $M$ probable classes $\mathcal{C} = [c_1, c_2, ..., c_M]$ and a sample $\mathbf{x}$ in $\mathbf{x}_s$, the $i-$th
485 client's logits output for $c_m$ is denoted as $f_{\boldsymbol{\theta}_t^{i,K}}(x)[m]$, and we get its score:

$$s_i = \sum_{\mathbf{x} \in \mathbf{x}_s} \sum_{c_m \in \mathcal{C}} \mathbb{1}(f_{\boldsymbol{\theta}_t^{i,K}}(\mathbf{x})[m] = \text{median}(F_{\boldsymbol{\theta}_t^K}(\mathbf{x})[m]))$$

$$\text{where } F_{\boldsymbol{\theta}_t^K}(\mathbf{x})[m] = [f_{\boldsymbol{\theta}_t^{1,K}}(\mathbf{x})[m], f_{\boldsymbol{\theta}_t^{2,K}}(\mathbf{x})[m], ..., f_{\boldsymbol{\theta}_t^{K,K}}(\mathbf{x})[m]]$$

487 Then FedRAD normalizes $s_i \leftarrow s_i / \sum_{i=1}^{K}(s_i)$ to let all the clients' scores adding up to 1, and
488 aggregates local models. The major difference between FedDF and FedRAD during distillation is that
489 the median $F_{\boldsymbol{\theta}_t^K}[m]$ is used instead of $\frac{1}{K} \sum_{i \in [K]} f_{\boldsymbol{\theta}_{t+1}^{i,K}}(\mathbf{x}_j)[m]$ to generate the $m$-th logit of teacher
490 model's soft labels.

491 **FedMV Pruning** [33] lets each client provide a ranking of all $p$ filters in its last convolutional layer
492 based on their averaged activation values and decide which filters would ultimately be pruned in the
493 global model. Suppose each client has a test dataset $\mathbf{x}_{\text{test}}$, and the activations of its last convolutional
494 layer is $\sigma(\mathbf{x}_{\text{test}}) = [\sigma_1(\mathbf{x}_{\text{test}}), \sigma_2(\mathbf{x}_{\text{test}}), ..., \sigma_p(\mathbf{x}_{\text{test}})]$. The $i$-th client get the ranks of the $p$ filter $\mathcal{R}^{i,K}$
495 based on the ascending order of $\sigma(\mathbf{x}_{\text{test}})^{i,K}$ (the smallest $\sigma(\cdot)$ means the smallest rank), and sends
496 $\mathcal{R}^{[i,K]}$ with its local model $\boldsymbol{\theta}^{[i,K]}$ for each round. The server averages the received rankings by filters
497 $\mathcal{R}^K = \text{avg}(\{\mathcal{R}^{i,K}\}_{1 \leq i \leq K})$ As a result, the server obtains a global ranking for all the filters in the
498 last convolutional layers.

499 **Bulyan**[9] argues that any current gradient aggregation rules (e.g. Krum, GeoMED, BRUTE), where
500 the aggregated vector is the result of a distance minimization scheme, can not defend the proposed
501 malicious updates that is highly-divergent on only a few coordinates while keeping the others closed.
502 In view of this, more than a combination of these rules, Bulyan bounds the aggregated updates around
503 the median of all proposed updates at each coordinate and excludes those potentially malicious
504 updates that disagree a lot. The two steps of Bulyan can be formalized as follows:

(1). Choose the updates closest to other updates among the proposed local updates. For pairwise distance, this would be the sum of Euclidean distance to other models. Move the chosen update from the "received set" to the "selection set", noted $\mathcal{S}$. Repeat the procedure until get $n-2f$ updates in $\mathcal{S}$.

(2). Aggregate $n-4f$ updates closest to the median by coordinate. Hence for each $i \in [1...d]$ (Suppose the uploaded models has $d$ dimensions), The resulting update $\boldsymbol{G} = \boldsymbol{\theta}_t - \boldsymbol{\theta}_{t-1} = (\boldsymbol{G}[1]...\boldsymbol{G}[d])$, so that for each of its coordinates $\boldsymbol{G}[\cdot]$:

$$\forall i \in [1,...,d], \; \boldsymbol{G}[i] = \frac{1}{n-4f} \sum_{\boldsymbol{X} \in \mathcal{M}[i]} \boldsymbol{X}[i]$$

$$\text{where} \quad \mathcal{M}[i] = \underset{\mathcal{R} \subset \mathcal{S}, |\mathcal{R}| = n-4f}{\arg\min} \left( \sum_{\boldsymbol{X} \in \mathcal{R}} |\boldsymbol{X}[i] - \text{median}[i]| \right) \tag{E.1}$$

$$\text{median}[i] = \underset{m = \boldsymbol{Y}_i, \boldsymbol{Y} \in \mathcal{S}}{\arg\min} \left( \sum_{\boldsymbol{Z} \in \mathcal{S}} |m - \boldsymbol{Z}_i| \right)$$

Simply stated: each i-th coordinate of $\boldsymbol{G}$ equals to the average of the $n-4f$ closest i-th coordinates to the median of the $n-2f$ selected updates.

**Robust LR** requires a sufficient number of proposed updates with the same signs to decide the global optimization direction. It assumes that the direction of proposed updates for benign clients and malicious clients is in most cases inconsistent, hence the presence of malicious clients would change the distribution of the signs of all proposed local updates. For each dimension with the sum of signs of updates fewer than a pre-defined threshold $\beta$, the learning rate is multiplied by $-1$. With the number of adversarial agents sufficiently below $\beta$, Robust LR is expected to move the global model from the backdoored model to the benign one. Since Robust LR only adjusts the learning rate, the approach is agnostic to the aggregation rules. For example, it can trivially work with update clipping and noise addition.

**DeepSight** [26] inspects the output probabilities $f_{\boldsymbol{\theta}^{i,K}}(\mathbf{x}_{\text{rand}}) \in \mathbb{R}^{d_0 \times d_c}$ ($d_c$ is the number of classes) of the i-th local model $\boldsymbol{\theta}^{i,K}$ on $d_0$ given random $d_1$-dimensional inputs $\mathbf{x}_{\text{rand}} \in \mathbb{R}^{d_0 \times d_1}$. After inspect the model's sample-wise average $(\bar{f}_{\boldsymbol{\theta}^{i,K}}(\boldsymbol{X}_{\text{rand}})) = \sum_{p=1}^{n_0} (f_{\boldsymbol{\theta}^{i,K}}(\boldsymbol{X}_{\text{rand}}[p])) \in \mathbb{R}^{d_c}$ and label the potentially malicious clients, DeepSight applies DBSCAN on participant clients [6] three times with distance matrices $D_{\text{bias}}, D_{\text{conv}}, D_{\text{prob}} \in \mathbb{R}^{K \times K}$ (assume all the local models have the same architecture with $L$ layers).

- $D_{\text{bias}}[i,j] = 1 - \text{cosine}(\boldsymbol{u}_{t+1}^{i,K,[L]}, \boldsymbol{u}_t^{j,K,[L]})$, $\boldsymbol{u}_{t+1}^{i,K,[L]}$ is the update of the i-th local update in their last layers.

- $D_{\text{conv}}[i,j] = ||\boldsymbol{w}_{t+1}^{i,K,[L]} - \boldsymbol{w}_t^{j,K,[L]}||$ is the Euclidean distance of the i-th and j-th local models' last layers $\boldsymbol{w}_{t+1}^{i,K,[L]}$ and $\boldsymbol{w}_t^{j,K,[L]}$

- $D_{\text{prob}}[i,j] = ||\bar{f}_{\boldsymbol{\theta}^{i,K}}(\mathbf{x}_{\text{rand}}) - \bar{f}_{\boldsymbol{\theta}^{j,K}}(\mathbf{x}_{\text{rand}})||$ is the Euclidean distance of clients' output probabilities for the global random vectors.

After get three clustering result vectors $Re_{\text{bias}}, Re_{\text{conv}}, Re_{\text{prob}} \in \mathbb{R}^K$ (For example, $Re_{\text{bias}}[i]$ is the i-th local model's cluster label based on $D_{\text{bias}}$), DeepSight defines a new distance matrix $D_{\text{final}}$ as:

$$D_{\text{final}}[i,j] = \sum_{r \in Res} \mathbb{1}(r[i] = r[j]) \text{ where } Res = \{Re_{\text{bias}}, Re_{\text{conv}}, Re_{\text{prob}}\}$$

DeepSight performs DBSCAN the last time according to $D_{\text{final}}[i,j]$ and get $Re_{\text{final}}$. Based on $Re_{\text{final}}$, the cluster with potentially malicious clients more than a given proportion threshold would be excluded from the next round of aggregation.

**CRFL**[34] clips the training-time global model parameters $Clip_{\rho_t}(\boldsymbol{\theta}_t) = \boldsymbol{\theta}_t / \max\left(1, \frac{||\boldsymbol{\theta}_t||}{\rho_t}\right)$ so that its norm is bounded by $\rho_t$, and then add isotropic Gaussian noise $\widetilde{\boldsymbol{\theta}}_t \leftarrow Clip_{\rho_t}(\boldsymbol{\theta}_t) + \epsilon_t$, where $\epsilon_t \sim \mathcal{N}(0, \sigma_t^2 \mathbf{I})$. Aligned with the training time Gaussian noise (perturbing), CRFL adopts the same Gaussian smoothing measures $\mu(\boldsymbol{\theta}) = \mathcal{N}(\boldsymbol{\theta}, \sigma_T^2 \mathbf{I})$ $M$ times independently on the tested model, to get $M$ sets of noisy model parameters, such that $\widetilde{\boldsymbol{\theta}}_T^k \leftarrow Clip_{\rho_t}(\boldsymbol{\theta}_t) + \epsilon_t^k$, runs the classifier with each set of noisy model parameters $\widetilde{\boldsymbol{\theta}}_T^k$ for one test sample $x_{test}$ to returns its class counts, with which take the voted most probable class and its probability.

# F  Additional Experimental Details

## F.1  General Experimental Settings

We evaluate the attacks on two classification datasets with non-i.i.d. data distributions: CIFAR-10 [15] and Tiny-ImageNet [16]. To simulate non-i.i.d. training data and supply the server with unbalanced samples from each class for model refinement, we divide the training images (in both datasets) using a Dirichlet distribution [23] with a concentration hyperparameter $h$ (a larger $h$ means a more i.i.d. data distribution). A shared global model is trained by all participants each round for aggregation. We evaluate CIFAR-10 and Tiny Imagenet dataset separately with a simple CNN (2 convolutional layers and 2 fully connected layers) and Resnet-18. Each participating client selected in one round will train for local epochs using SGD with the learning rate of $\eta = 0.001$ for both CIFAR and Tiny ImageNet. To ensure that the backdoor trigger is practical and hard to notice by human eyes, we limit the size of the trigger to a small $3 \times 3$ square on the CIFAR-10 dataset, and $4 \times 4$ for the Tiny ImageNet dataset. For F3BA, we set the trigger optimization iteration $P = 10$ and $\eta = 0.1$. We apply different candidate parameter selection porportions $1\%$ and $0.1\%$ respectively for convolutional layers and fully-connected layers.

## F.2  Compare with the DBA's Experimental Setting

Our goal is to use a realistic experimental settings to fairly and accurately evaluate the real-world performance of various federated backdoor defenses in the face of advanced backdoor attacks. We believe that it is unrealistic to specify a fixed number of malicious clients in each training round as in the experimental setup of DBA. In reality, due to the server's lack of knowledge on the clients (the server cannot know in advance whether a client is benign or malicious), it can only randomly select a subset of clients to participate in each training round (the number of malicious clients is unknown). In this case, it is more practical to randomly select participating clients among all the clients without distinguishing between benign and malicious clients as the experimental setting in F3BA.

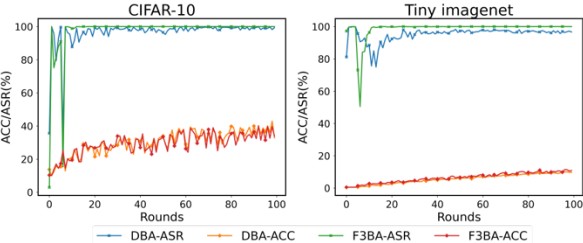

Figure 8: Attack success rate and accuracy of F3BA and DBA against plain FedAvg on CIFAR-10 and Tiny-ImageNet datasets under the setting of DBA.

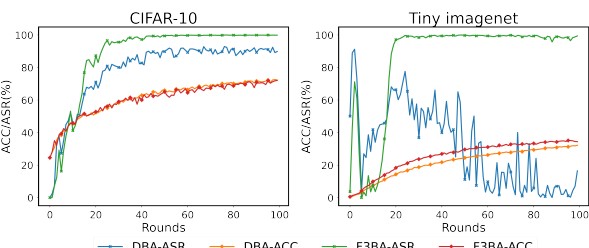

Figure 9: Attack success rate and accuracy of F3BA and DBA against plain FedAvg on CIFAR-10 and Tiny-ImageNet datasets under the setting of F3BA.

We attack plain FedAvg with F3BA and DBA in the pre-tuned (in DBA) and post-tuned (in F3BA) experimental settings. Figure.8 and Figure.9 show that even without any defense, DBA could not evade FedAvg on Tiny-Imagenet dataset in the post-tuned setting, while F3BA succeed in both datasets for the two settings. Our chosen setting is actually harder and more practical than the setting

of DBA. Since all the clients are randomly selected, as Figure.10 shows, the proportion of malicious clients among all the participating clients is much less than that in DBA's setting.

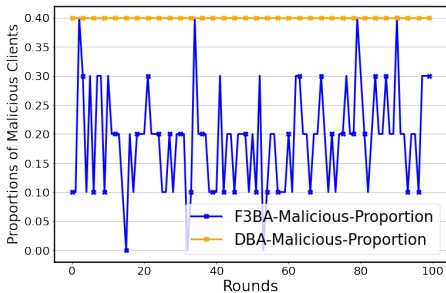

Figure 10: The proportion of malicious clients of each training round in DBA's and F3BA's settings.

### F.3 Additional Details on the Defense Baselines

**FedDF** [20] leverages unlabeled data or artificially generated samples from a GAN's generator to achieve robust server-side model fusion, aggregate knowledge from all received (heterogeneous) client models. Formally, in the $t$-th round, the server first obtains a global model $\boldsymbol{\theta}_{t+1}$ with standard FedAvg as equation 2.1

FedDF distills the global model with the ensembled logit predictions on clean unlabeled samples from each client's model, hence overcoming the limitation of the number of data samples and eliminating the effect of data drift.

**FedRAD** [29] utilizes median-based scoring along with knowledge distillation for ensemble distillation. In the $t$-th round with $K$ participating clients, the median-based scoring assigns the $i$-th client a score $s_i$ by counting how many times it gave the prediction for a server-side dataset $\mathbf{x}_s$. For a classification task with $M$ probable classes $\mathcal{C} = [c_1, c_2, ..., c_M]$ and a sample $\mathbf{x}$ in $\mathbf{x}_s$, the $i-$th client's logits output for $c_m$ is denoted as $f_{\boldsymbol{\theta}_t^{i,K}}(x)[m]$, and we get its score:

$$s_i = \sum_{\mathbf{x} \in \mathbf{x}_s} \sum_{c_m \in \mathcal{C}} \mathbb{1}(f_{\boldsymbol{\theta}_t^{i,K}}(\mathbf{x})[m] = \text{median}(F_{\boldsymbol{\theta}_t^K}(\mathbf{x})[m]))$$
$$\text{where } F_{\boldsymbol{\theta}_t^K}(\mathbf{x})[m] = [f_{\boldsymbol{\theta}_t^{1,K}}(\mathbf{x})[m], ..., f_{\boldsymbol{\theta}_t^{K,K}}(\mathbf{x})[m]]$$

Then FedRAD normalizes $s_i \leftarrow s_i / \sum_{i=1}^K (s_i)$ to let all the clients' scores adding up to 1, and aggregates local models. The major difference between FedDF and FedRAD during distillation is that the median $F_{\boldsymbol{\theta}_t^K}[m]$ is used instead of $\frac{1}{K} \sum_{i \in [K]} f_{\boldsymbol{\theta}_{t+1}^{i,K}}(\mathbf{x}_j)[m]$ to generate the $m$-th logit of teacher model's soft labels.

**FedMV Pruning** [33] lets each client provide a ranking of all $p$ filters in its last convolutional layer based on their averaged activation values and decide which filters would ultimately be pruned in the global model. Suppose each client has a test dataset $\mathbf{x}_{\text{test}}$, and the activations of its last convolutional layer is $\sigma(\mathbf{x}_{\text{test}}) = [\sigma_1(\mathbf{x}_{\text{test}}), \sigma_2(\mathbf{x}_{\text{test}}), ..., \sigma_p(\mathbf{x}_{\text{test}})]$. The $i$-th client get the ranks of the $p$ filter $\mathcal{R}^{i,K}$ based on the ascending order of $\sigma(\mathbf{x}_{\text{test}})^{i,K}$ (the smallest $\sigma(\cdot)$ means the smallest rank), and sends $\mathcal{R}^{[i,K]}$ with its local model $\boldsymbol{\theta}^{[i,K]}$ for each round. The server averages the received rankings by filters $\mathcal{R}^K = \text{avg}(\{\mathcal{R}^{i,K}\}_{1 \leq i \leq K})$. As a result, the server obtains a global ranking for all the filters in the last convolutional layers.

**Bulyan** [9] argues that any current gradient aggregation rules (e.g. Krum, GeoMED, BRUTE), where the aggregated vector is the result of a distance minimization scheme, can not defend the proposed malicious updates that is highly-divergent on only a few coordinates while keeping the others closed. In view of this, more than a combination of these rules, Bulyan bounds the aggregated updates around the median of all proposed updates at each coordinate and excludes those potentially malicious updates that disagree a lot. The two steps of Bulyan can be formalized as follows:

(1). Choose the updates closest to other updates among the proposed local updates. For pairwise distance, this would be the sum of Euclidean distance to other models. Move the chosen update from the "received set" to the "selection set", noted $\mathcal{S}$. Repeat the procedure until get $n - 2f$ updates in $\mathcal{S}$.

(2). Aggregate $n-4f$ updates closest to the median by coordinate. Hence for each $i \in [1...d]$ (Suppose the uploaded models has $d$ dimensions), The resulting update $\boldsymbol{G} = \boldsymbol{\theta}_t - \boldsymbol{\theta}_{t-1} = (\boldsymbol{G}[1]...\boldsymbol{G}[d])$, so that for each of its coordinates $\boldsymbol{G}[\cdot]$:

$$\forall i \in [1,...,d], \ \boldsymbol{G}[i] = \frac{1}{n-4f} \sum_{\boldsymbol{X} \in \mathcal{M}[i]} \boldsymbol{X}[i]$$

$$\text{where} \quad \mathcal{M}[i] = \underset{\mathcal{R} \subset \mathcal{S}, |\mathcal{R}| = n-4f}{\arg\min} \left( \sum_{\boldsymbol{X} \in \mathcal{R}} |\boldsymbol{X}[i] - \text{median}[i]| \right) \tag{F.1}$$

$$\text{median}[i] = \underset{m=\boldsymbol{Y}_i, \boldsymbol{Y} \in \mathcal{S}}{\arg\min} \left( \sum_{\boldsymbol{Z} \in \mathcal{S}} |m - \boldsymbol{Z}_i| \right)$$

Simply stated: each i-th coordinate of $\boldsymbol{G}$ equals to the average of the $n-4f$ closest i-th coordinates to the median of the $n-2f$ selected updates.

**Robust LR** requires a sufficient number of proposed updates with the same signs to decide the global optimization direction. It assumes that the direction of proposed updates for benign clients and malicious clients is in most cases inconsistent, hence the presence of malicious clients would change the distribution of the signs of all proposed local updates. For each dimension with the sum of signs of updates fewer than a pre-defined threshold $\beta$, the learning rate is multiplied by $-1$. With the number of adversarial agents sufficiently below $\beta$, Robust LR is expected to move the global model from the backdoored model to the benign one. Since Robust LR only adjusts the learning rate, the approach is agnostic to the aggregation rules. For example, it can trivially work with update clipping and noise addition.

**DeepSight** [26] inspects the output probabilities $f_{\boldsymbol{\theta}^{i,K}}(\mathbf{x}_{\text{rand}}) \in \mathbb{R}^{d_0 \times d_c}$ ($d_c$ is the number of classes) of the $i$-th local model $\boldsymbol{\theta}^{i,K}$ on $d_0$ given random $d_1$-dimensional inputs $\mathbf{x}_{\text{rand}} \in \mathbb{R}^{d_0 \times d_1}$. After inspect the model's sample-wise average $(\bar{f}_{\boldsymbol{\theta}^{i,K}}(\boldsymbol{X}_{\text{rand}})) = \sum_{p=1}^{n_0}(f_{\boldsymbol{\theta}^{i,K}}(\boldsymbol{X}_{\text{rand}}[p])) \in \mathbb{R}^{d_c}$ and label the potentially malicious clients, DeepSight applies DBSCAN on participant clients [6] three times with distance matrices $D_{\text{bias}}, D_{\text{conv}}, D_{\text{prob}} \in \mathbb{R}^{K \times K}$ (assume all the local models have the same architecture with $L$ layers).

- $D_{\text{bias}}[i,j] = 1 - \text{cosine}(\boldsymbol{u}_{t+1}^{i,K,[L]}, \boldsymbol{u}_t^{j,K,[L]})$, $\boldsymbol{u}_{t+1}^{i,K,[L]}$ is the update of the $i$-th local update in their last layers.

- $D_{\text{conv}}[i,j] = ||\boldsymbol{w}_{t+1}^{i,K,[L]} - \boldsymbol{w}_t^{j,K,[L]}||$ is the Euclidean distance of the $i$-th and $j$-th local models' last layers $\boldsymbol{w}_{t+1}^{i,K,[L]}$ and $\boldsymbol{w}_t^{j,K,[L]}$

- $D_{\text{prob}}[i,j] = ||\bar{f}_{\boldsymbol{\theta}^{i,K}}(\mathbf{x}_{\text{rand}}) - \bar{f}_{\boldsymbol{\theta}^{j,K}}(\mathbf{x}_{\text{rand}})||$ is the Euclidean distance of clients' output probabilities for the global random vectors.

After getting the three clustering result vectors $Re_{\text{bias}}, Re_{\text{conv}}, Re_{\text{prob}} \in \mathbb{R}^K$ (For example, $Re_{\text{bias}}[i]$ is the $i$-th local model's cluster label based on $D_{\text{bias}}$), DeepSight defines a new distance matrix $D_{\text{final}}$ as:

$$D_{\text{final}}[i,j] = \sum_{r \in Res} \mathbb{1}(r[i] = r[j]), \tag{F.2}$$

$$\text{where } Res = \{Re_{\text{bias}}, Re_{\text{conv}}, Re_{\text{prob}}\}$$

DeepSight performs DBSCAN the last time according to $D_{\text{final}}[i,j]$ and get $Re_{\text{final}}$. Based on $Re_{\text{final}}$, the cluster with potentially malicious clients more than a given proportion threshold would be excluded from the next round of aggregation.

**CRFL** [34] clips the training-time global model parameters $Clip_{\rho_t}(\boldsymbol{\theta}_t) = \boldsymbol{\theta}_t / \max\left(1, \frac{||\boldsymbol{\theta}_t||}{\rho_t}\right)$ so that its norm is bounded by $\rho_t$, and then add isotropic Gaussian noise $\widetilde{\boldsymbol{\theta}}_t \leftarrow Clip_{\rho_t}(\boldsymbol{\theta}_t) + \epsilon_t$, where $\epsilon_t \sim \mathcal{N}(0, \sigma_t^2 \mathbf{I})$. Aligned with the training time Gaussian noise (perturbing), CRFL adopts the same Gaussian smoothing measures $\mu(\boldsymbol{\theta}) = \mathcal{N}(\boldsymbol{\theta}, \sigma_T^2 \mathbf{I})$ $M$ times independently on the tested model, to get $M$ sets of noisy model parameters, such that $\widetilde{\boldsymbol{\theta}}_T^k \leftarrow Clip_{\rho_t}(\boldsymbol{\theta}_t) + \epsilon_t^k$, runs the classifier with each set of noisy model parameters $\widetilde{\boldsymbol{\theta}}_T^k$ for one test sample $x_{test}$ to returns its class counts, with which take the voted most probable class and its probability.

# G Ablation Study

In this section, we provide ablation studies towards our proposed F3BA method and study how various factors affect the ASR and ACC of our proposed attack. To eliminate the influence of random client participation, in this, section, we set a total of 10 participant clients with only 1 malicious client, and all the participants would be selected in each round. The data heterogeneity hyperparameter $h = 1.0$ (Except when we investigate the effects of data heterogeneity). We do not set more clients or lower data heterogeneity because it would make it easier to achieve high ASR for F3BA with various parameter settings, which is not conducive for us to investigate the effects of various factors on the backdoor attack.

## G.1 Attack with Different Trigger

In our attack design, the malicious clients do not need to share the optimized trigger during training, instead, they optimize their own triggers when conducting the F3BA attack. During the test phase, we simply average all the optimized triggers as a global trigger and attached it to the test dataset for ASR evaluation. Note that even if we do not perform averaging but directly use one of the optimized triggers, the attack still works.

Table.1 shows the ASR of different triggers on CIFAR-10 and Tiny-Imagenet datasets. Note that whether using the average or local triggers does not have a major impact on the performance of F3BA on both datasets.

| Tested Tigger | Round | CIFAR-10 | Tiny-imagenet |
|---|---|---|---|
| Averaged Trigger | 50 | 97.97% | 97.01% |
| | 100 | 99.23% | 99.15% |
| Local Trigger #1 | 50 | 98.03% | 96.36% |
| | 100 | 99.49% | 99.11% |
| Local Trigger #2 | 50 | 98.57% | 96.57% |
| | 100 | 99.49% | 98.95% |
| Local Trigger #3 | 50 | 97.87% | 98.15% |
| | 100 | 98.35% | 99.62% |
| Local Trigger #4 | 50 | 98.84% | 96.57% |
| | 100 | 98.41% | 98.48% |

Table 1: ASR of F3BA and DBA attacks with the averaged global trigger and clients' local triggers against plain FedAvg.

## G.2 Attack Other Network Architecture

As mentioned in Section.B, the Focused Flip operations can be generally applied to any network structure that rely on the dot product, and be applied independently to boost the traditional training-based backdoor attack. We show the ASR/ACC of F3BA on attacking plain FedAvg with MLP models to verify the applicability of the F3BA attack on architectures beyond CNN. Table.2 suggests that F3BA is still highly effective on MLPs (and still better than the DBA baseline) without loss on the performance of its main task.

| Round | CIFAR-ACC | | CIFAR-ASR | | TinyImagenet-ACC | | TinyImagenet-ASR | |
|---|---|---|---|---|---|---|---|---|
| | F3BA | DBA | F3BA | DBA | F3BA | DBA | F3BA | DBA |
| 25 | 44.13% | 44.57% | 98.76% | 87.21% | 7.06% | 7.02% | 98.52% | 89.54% |
| 50 | 47.17% | 47.26% | 99.78% | 93.30% | 8.58% | 8.87% | 97.37% | 86.35% |
| 75 | 48.53% | 49.13% | 99.82% | 91.55% | 9.40% | 9.50% | 98.39% | 90.39% |
| 100 | 51.10% | 50.99% | 99.88% | 93.68% | 10.04% | 10.00% | 98.94% | 93.54% |

Table 2: ASR/ACC of F3BA and DBA with MLP network architecture against plain FedAvg.

### G.3 Attack with More Benign Clients

We test F3BA on FedAvg with in total 100 clients, of which 4 are malicious. 40 clients are randomly selected for each round. When the number of malicious clients is fixed, the benign clients becomes more thus to some extent deterring the attack from both F3BA and DBA. As Table.3, we can observe that the advantages of F3BA over DBA still holds. When all the clients are randomly selected, the decreasing chances for malicious clients being selected do partially slows down the process of F3BA's evasion but not able to remove the injected backdoor. In this circumstance, F3BA still performs better than DBA. Stopping F3BA entirely by the number of benign clients would require potentially much more benign clients and fewer malicious clients.

| Round | CIFAR-ACC | | CIFAR-ASR | | TinyImagenet -ACC | | TinyImagenet -ASR | |
|---|---|---|---|---|---|---|---|---|
| | F3BA | DBA | F3BA | DBA | F3BA | DBA | F3BA | DBA |
| 25 | 39.01% | 38.10% | 12.39% | 10.12% | 9.25% | 9.64% | 7.15% | 4.72% |
| 50 | 46.54% | 45.93% | 34.26% | 20.27% | 17.50% | 17.23% | 18.10% | 9.27% |
| 75 | 51.21% | 50.95% | 56.20% | 25.55% | 20.05% | 20.66% | 42.06% | 15.20% |
| 100 | 55.38% | 55.60% | 75.25% | 24.36% | 23.84% | 24.20% | 60.11% | 30.05% |

Table 3: ASR/ACC with more benign clients against plain FedAvg. 100 clients, 4 malicious.

### G.4 Attack Sparsification-based Defense

Besides 3 Model Refinement defenses, 3 Robust Aggregation defenses, and 1 Certified Robustness defenses in Section.3,we also explore the effect of SparseFed, a theoretical framework for analyzing the robustness of defenses against poisoning attacks. Since it is not specifically designed for the robustness towards backdoor attacks, we put the result in the supplementary as Table.4.

Based on the model architecture and task complexity, we set the number of accepted parameters $K = 1e4$ for CNN and $K = 4e5$ for ResNet-18 respectively. From the Table.4, we can observe that although SparseFed only allows a small fraction of aggregated parameters for global model updates at each round, it still can be backdoored by our F3BA.

| Round | CIFAR | | TinyImagenet | |
|---|---|---|---|---|
| | ACC | ASR | ACC | ASR |
| 25 | 44.70% | 76.75% | 1.75% | 9.64% |
| 50 | 49.63% | 90.51% | 4.78% | 49.56% |
| 75 | 54.43% | 99.30% | 8.05% | 87.21% |
| 100 | 57.71% | 99.75% | 11.10% | 82.77% |

Table 4: ASR/ACC with against sparseFed FedAvg. 20 clients, 4 malicious.

We conjecture that the sparsity criterion cannot rule out all the backdoor-related model parameters as our attack does not necessarily lead to an update that is small in magnitude.

### G.5 Effect of Data Heterogeneity

We study how data heterogeneity would affect the attack of F3BA. We manually adjust the concentration hyperparameter $h$ to split non-i.i.d dataset.(The larger the $h$, the more i.i.d the data is distributed). On the two datasets, the ASR both grows when the $h$ becomes smaller, and the ACC decreases at the same time. The result shows that lower $h$ strongly hurts the accuracy of the global model but gives convenience to the backdoor attack.

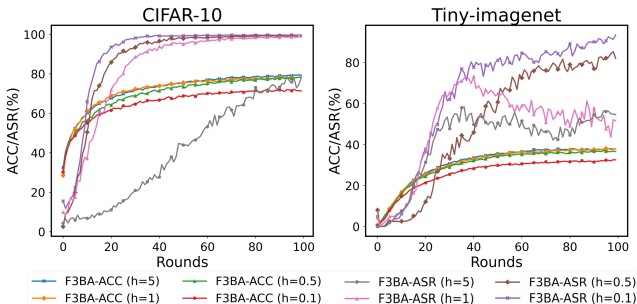

Figure 11: Attack success rate and accuracy against FedAvg under different data heterogeneity $h$.

### G.6 Effect of The Proportion of Candidate Parameters

In our experiments, we find that different proportions of candidate parameters should be used for the different parts of a neural network in order to achieve a better attack. To valid this, we conducted a grid search on CIFAR-10 for ACC and ASR at the $20^{th}$ round. Based on the result in Figure 12, moderately scaling up the candidate parameters rate (e.g. from 0.02 to 0.05) in fully-connected layers would increase the ASR without the loss of ACC. Meanwhile, too high candidate parameters proportion for fully-connected layers can cause an obvious loss of ASR. Intuitively, the parameters of the convolutional part are more sparse than the fully-connected part and therefore can be selected with a higher candidate parameter proportion. Similarly, fully-connected layers indeed involve more parameters, and thus only require a smaller candidate parameter proportion. According to our practice, a sound choice for attackers is to set the proportions for convolutional layers and fully-connected layers respectively below 5% and 1%.

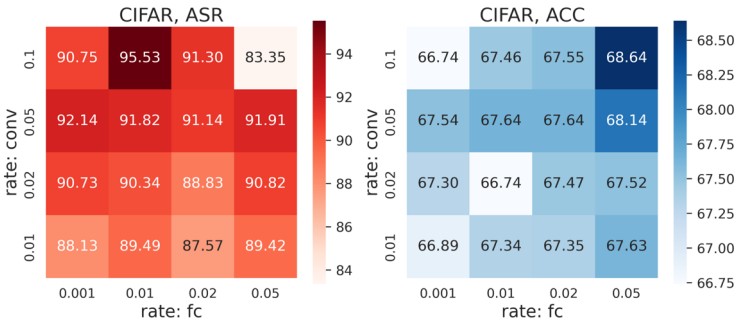

Figure 12: Attack success rate and accuracy with different candidate parameter proportion for convolutional layers and fully-connected layers on CIFAR-10 dataset.

### G.7 Effect of Local Training

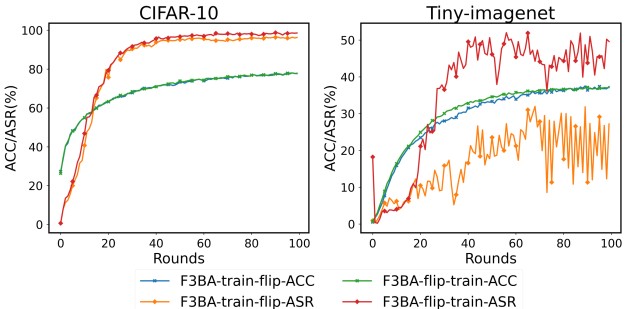

Figure 13: Attack success rate and accuracy against FedAvg under different data heterogeneity $h$.

As discussed in Section 2.2, local training is still a must for the effects of F3BA, while simply flipping without training can not induce activation led by the trigger to the targeted label. We further discover how the order of local training and Focused Flip would affect our attack. Based on the results on CIFAR-10 and Tiny Imagenet, the flipping-training-pipeline can achieve better ASR than the flip-training one. It also ensures a slightly higher ACC. The significant difference of ASR on attacking Tiny Imagenet dataset for two pipelines also suggests that training a local model to bridge the sudden changes in model weights caused by focused flip can be of benefit to the effectiveness of the attack.

