# OpenReview forum: "On the Vulnerability of Backdoor Defenses for Federated Learning"
_NeurIPS.cc/2022/Workshop/Federated_Learning — FL-NeurIPS 2022 Poster_

### Official Review · Reviewer_6jxr · 2022-10-16

Summary

The authors propose a more persistent and stealthy backdoor attack. Specifically, the attack selectively flips the signs of a small proportion of network weights and jointly optimizes the trigger pattern with the model.
Experiments show the effectiveness of the attack against the defenses from three major categories, and the authors give takeaways guidelines for the choice of backdoor defenses for different settings.



Strength
1. A novel approach for the backdoor attack in FL based on parameter selection, sign flipping, and trigger optimization.
2. Comprehensive evaluations of the defenses from three main categories show that the proposed attack is more effective than DBA.


Weakness
1. Comparison to other FL attacks [1,2,3] might further strengthen the submission.

References

[1] Eugene Bagdasaryan, Andreas Veit, Yiqing Hua, Deborah Estrin, and Vitaly Shmatikov. How to backdoor federated learning. In International Conference on Artificial Intelligence and Statistics, pages 2938–2948. PMLR, 2020.

[2] Hongyi Wang, Kartik Sreenivasan, Shashank Rajput, Harit Vishwakarma, Saurabh Agarwal, Jy-yong Sohn, Kangwook Lee, and Dimitris Papailiopoulos. Attack of the tails: Yes, you really can backdoor federated learning. arXiv preprint arXiv:2007.05084, 2020

[3] Minghong Fang, Xiaoyu Cao, Jinyuan Jia, and Neil Gong. Local model poisoning attacks to { Byzantine-Robust } federated learning. In 29th USENIX Security Symposium (USENIX Security 20), pages 1605–1622, 2020.

---

### Official Review · Reviewer_7mei · 2022-10-17
**Cool idea but limited novelty**

Backdoor defenses in federated learning aim to change the model output of specific data points when a specific trigger is presented, by manipulating some client updates during the federated training process. This work proposes a new federated backdoor attack framework that combines targeted sign flips of model weights with joint optimization of the trigger pattern with client model. Experiments on the CIFAR-10 and TinyNet image dataset show that the resulting attack algorithm, F3BA, can defeat well-known model refinement, robust aggregation, and certified robustness defenses against backdoor attacks. Existing backdoor attack methods, such as DBA, demonstrate less effectiveness against these defenses.

+ The proposed defense method seems interesting and, if not entirely novel, fairly effective. F3BA consistently achieves better attack success rates than DBA in the experiments, and usually the attack success rate is higher than the test accuracy.

--In general, it would be helpful for the paper to provide more intuitions behind the proposed attack methods and why they would work. For example, how is the trigger determined? Would it be optimizable in practice, or could such optimization compromise the attack effectiveness (e.g., is the specific trigger chosen based on the desired model vulnerability, for which optimizing the trigger would requiring changing the overall goal of the attacker)? How do the design of trigger optimization and focused flipping help to evade the various types of defenses?

--It is not clear how the selection of candidate parameters for manipulation differs from that proposed for prior backdoor attacks in federated learning. Is the main difference the use of the importance score for different weights?

--The list of contributions claims that F3BA can be universally applied to model architectures beyond convolutional neural networks, but the discussion seems to restrict itself to neural networks.

--It would be interesting to see ablation experiments evaluating the focused flipping and trigger optimization components of F3BA separately, to see which is contributing the most to F3BA’s attack success.

--The guidelines of which backdoor defense method to use seem quite straightforward and intuitive. It would be more useful to suggest defense methods against the new F3BA attack.

--How many malicious clients were used in the experiments? This would seem to greatly affect the success of F3BA and any backdoor attack in general.

---

### Official Review · Reviewer_Moz7 · 2022-10-17
**A decent contribution to robust federated learning**

This paper proposes F3BA a carefully designed backdoor attack for better persistence and harder for being detected. The paper also studies various defense methods against F3BA via extensive experiments. The results generally demonstrate all categories of defense methods have their pros and cons. And F3BA injects the backdoor attack effectively and the attack does persist.

Pros:
- The paper is well written.
- Enhancing the robustness of federated learning is a promising direction to pursue.
- The experimental results are extensive.

Cons:
- The threat model seems to be a bit strong, i.e., allowing the attacker to control the entire training procedure. Imagine, in the App Google Board, users do not seem to be able to control the training. The only thing an attacker can control is their local data.
- The computation overhead of the attack is not explicitly discussed.

---

### Decision · Program_Chairs · 2022-10-20

Accept (Poster)